# Imitation from Observation With Bootstrapped Contrastive Learning

**Medric B. Djeafea Sonwa**
University of Montreal and Mila
`medric.sonwa@umontreal.ca`

**Johanna Hansen**
McGill University and Mila
`johanna.hansen@mail.mcgill.ca`

**Eugene Belilovsky**
Concordia University and Mila
`eugene.belilovsky@concordia.ca`

## Abstract

Imitation from observation (IfO) is a learning paradigm that consists of training autonomous agents in a Markov Decision Process (MDP) by observing expert demonstrations without access to its actions. These demonstrations could be sequences of environment states or raw visual observations of the environment. Recent work in IfO has focused on this problem in the case of observations of low-dimensional environment states, however, access to these highly-specific observations is unlikely in practice. In this paper, we adopt a challenging, but more realistic problem formulation, learning control policies that operate on a learned latent space with *access only to visual demonstrations* of an expert completing a task. We present BootIfOL, an IfO algorithm that aims to learn a reward function that takes an agent trajectory and compares it to an expert, providing rewards based on similarity to agent behavior and implicit goal. We consider this reward function to be a distance metric between trajectories of agent behavior and learn it via contrastive learning. The contrastive learning objective aims to closely represent expert trajectories and to distance them from non-expert trajectories. The set of non-expert trajectories used in contrastive learning is made progressively more complex by bootstrapping from roll-outs of the agent learned through RL using the current reward function. We evaluate our approach on a variety of control tasks showing that we can train effective policies using a limited number of demonstrative trajectories, greatly improving on prior approaches that consider raw observations.

## 1 Introduction

Imitation from Observation (IfO) [24; 17] involves learning a policy function for an agent to solve a task using a set of demonstrations of an expert performing the same task. Distinct from Imitation Learning (IL) [12; 20; 18; 2], in IfO, the demonstrations are only sequences of observations of the environment by the expert. Depending on the environment and the information available to the agent, an observation can be a description of the environment at a given time (joint angles, distance between objects, direction vector coordinates, velocity, etc.) provided in the form of a low-dimensional state vector of the environment, or a raw visual of the environment subject to further analysis and processing. The advantage of the IfO formalization is that it allows more natural agent learning scenarios, where difficult-to-acquire precise action data is not available. Furthermore, it more accurately approximates the way in which humans imitate experts during learning. The two main methods in the literature that have been used to train IfO problem agents are adversarial methods and reward learning methods [27]. The adversarial methods [25; 26] adapt ideas from

Offline Reinforcement Learning Workshop at Neural Information Processing Systems, 2022

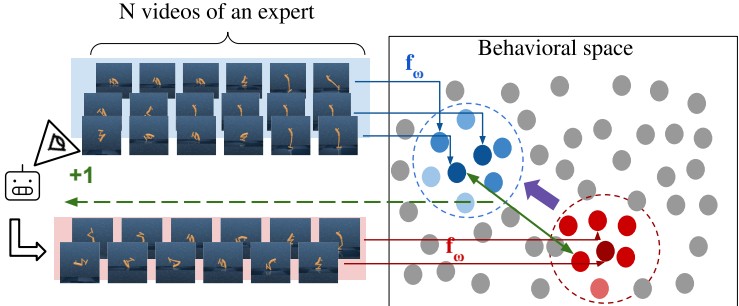

Figure 1: Illustration of our method. An encoder that takes visual demonstrations of agent trajectories and embeds them in a "behavioral space" that is trained using contrastive learning. This contrastive objective encourages successful trajectories to be near each other in latent space. We use this to encode N expert trajectories in a region of the behavioral space depicted in blue. The reward function corresponds to the distance of the agent's trajectory to the set of expert trajectories. As the agent progresses, its current trajectories are incorporated as "negative" examples for contrast indicated in red.

inverse reinforcement learning [10], proposing a GAN-like architecture [6] where the agent is a trajectory generator associated with a discriminator function that evaluates how similar the state transitions produced are to that of an expert.

These methods have proven their effectiveness in the case of low-dimensional state observations. When dealing with high-dimensional visuals, [25] uses a small stack (of size 3) of successive images to evaluate fake and real data. However, when using high-dimensional visual observations instead of low-dimensional state sequences, each video transcribes the variations of the system states without clearly designating the main and critical features that describe the environment. The fact of reducing the problem to a classification of small sequences limits the capacity of the discriminator to analyze and identifies these features describing optimal actions. Being able to correctly understand and represent the system state using environment images is essential to evaluate state transitions. Moreover, this approach limits the understanding of specific behaviors planning their actions over long horizons. On the other hand, reward learning methods [14; 17; 3] are based on the training of a reward function that will be subsequently used to reward an agent while applying a traditional RL algorithm. The main interest of such an approach is to learn a reward function that measures the efficiency of the agent actions based on the expert demonstrations. The different reward learning methods exploit a precise self-supervised learning objective, such as context variation between many executions [17], which allows the model to converge to a meaningful and efficient reward function for training the imitating agent.

In this paper, we present *BootIfOL*, a novel reward learning IfO algorithm that relies on the representation of agent behaviors in a latent space using raw pixel-based demonstrations, illustrated in Figure 1. Unlike adversarial approaches, we use the entire observation sequences with a Long Short-Term Memory (LSTM) neural network [11] to keep track of the agent's actions and to represent the induced behavior of the agent at each sequence timestep. This method directly exploits a limited set of visual demonstrations from an expert to train an imitating agent, without having access to the expert policy or actions. Inspired by Berseth *et al.* [1], we also train the reward function progressively with new visual trajectories generated by the agent. In contrast to [1], our method trains the trajectory encoding function on a set of failure trajectories before the agent training starts. This helps to ensure the consistency of the reward function from the beginning of the agent training. We also use Contrastive Multiview Coding [23] and Dense Predictive Coding [8] methods for self-supervised learning of trajectories. Like [17; 1], a reward function is trained by interacting with the environment to encourage the imitating agent's behavior to be similar to the behavior of the expert. The motivation behind this approach is: (1) To learn and estimate, at each timestep, the system state using the agent's visual observations (knowing that the real system state is not accessible); (2) To identify the behavior induced by the agent over a certain period of task execution and make sure that this behavior serves the same purpose as the expert. Our method also demonstrates the interest in a first-round training of the trajectory encoding function in order to provide meaningful rewards to the agent from the beginning of its training, contrary to other similar reward learning methods that bypass this step. Our method successfully solves a range of tasks in the Deepmind Control Suite [22] and the Meta-world environment [30], approaching episodic rewards of task experts.

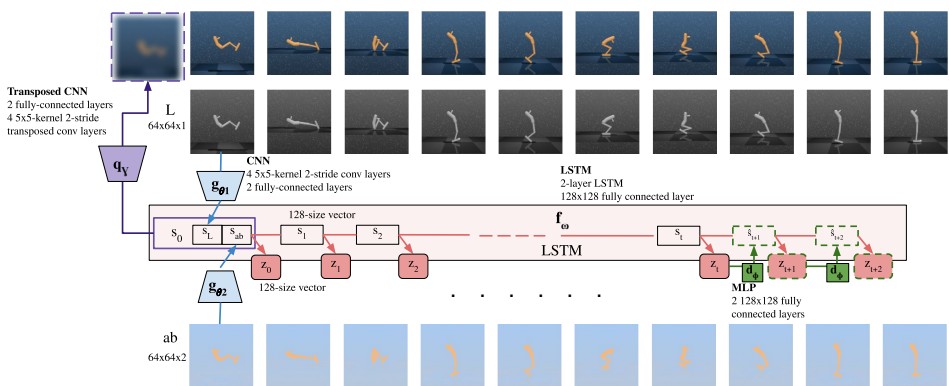

Figure 2: Training architecture of the imitation functions. For each episode, the video is decomposed in the Lab color space, constituting the L and ab views. Each frame is encoded by $g_\theta$ and decoded by $q_\gamma$. The resulting state sequence $s_0, ..., s_t$ is encoded using the LSTM $f_\omega$ to provide the sequence encoding $z_t$. $z_t$ is then processed by $d_\phi$ and $f_\omega$ to predict future image encodings.

## 2 Method

**Agent training** We aim to train an agent to perform a task only using a set of videos $\mathcal{D}_e$ of an expert performing the same task many times. Our proposal is based on the trajectory matching objective, which consists of training the agent to have a trajectory similar to that of the expert. In other words, the only information we seek to extract from the expert's videos is its behavior. Our algorithm is divided into two phases, an *Alignment Phase* and an *Interactive Phase*. During the Alignment Phase, we learn a sequence encoding function $f_\omega$, which encodes sequence of frames of an agent trajectory, and an associated distance metric between two sequences of two agent trajectories. We also jointly train an image encoding function $g_\theta$. Note that this image encoding function $g_\theta$ can also be replaced by a pre-trained encoder. We discuss the impact of this replacement in Section 3. In the Alignment Phase, we only use trajectories coming from two distributions: the distribution of expert's trajectories $O \sim p(\mathcal{D}_e)$, and a distribution of trajectories $O \sim p(\mathcal{D}_a)$ generated by a randomly defined policy function. During the Interactive Phase, we learn the agent policy $\pi$ using a standard Reinforcement Learning (RL) algorithm [21]. To obtain the reward we use the learned distance (trained during the Alignment Phase) between sequence encodings of the expert and the current agent. Critically, we fine-tune the image and sequence encoding functions with additional visual observations from our online interactions with the environment. In this work, we use DrQ-v2 [29] RL algorithm to train the agent policy, $\pi$, and its associated q-value function, $Q$, though our method is agnostic to the choice of the RL algorithm. Specifically, in the Interactive Phase at each new episode of the agent, we sample an expert episode $O_e \sim p(\mathcal{D}_e)$ that acts as a reference expert policy. At each agent step, $t$, we evaluate the distance between $o_{0:t}$, the agent's trajectory until the step $t$, and $o_{e,0:t}$ the expert's trajectory until $t$ by $d(o_{0:t}, o_{e,0:t}) = ||f_\omega(g_\theta(o_{0:t})) - f_\omega(g_\theta(o_{e,0:t}))||$. The reward assigned to the agent at this step is finally $r_t = -d(o_{0:t}, o_{e,0:t})$. The function $f_\omega$ from the Alignment Phase is trained only on expert trajectories and random trajectories, thus as the policy of the agent improves, the distribution of agent trajectories will change, making the distances produced inconsistent due to the shift of the trajectory distribution. In order to fix this in the Interactive Phase, we continue to update $f_\omega$ and $g_\theta$ on the new trajectories generated by the agent during the training of $\pi$. In Paragraph 3, we discuss the necessity of the Alignment Phase before training the agent. The full training process is detailed in Algorithm 1.

**Image encoding** We aim to encode videos (observation trajectories) and thus we need to find encodings of individual frames. Although a pre-trained image model may be used, a suitable one is not always available or appropriate (e.g. too expensive for the task). Thus we propose and evaluate learning the image encodings along with the sequence encodings. The image level objective contains three terms: (1) $\mathcal{L}_{triplet}$ which enforces image encodings similarity of adjacent frames, (2) $\mathcal{L}_{ae}$ which permits the encoding to be decoded back to an image, and finally (3) we learn a contrastive multiview encoding following [23] based on a Lab color space [4] image representation. The first term $\mathcal{L}_{triplet}$ compares the distance between an anchor image $o$, a positive image $o_p$ temporally close to the anchor, and a negative image $o_n$ distant from the anchor image in the sequence:

$$\mathcal{L}_{triplet}(\theta) = ||s - s_p||^2 + \max(\rho - ||s - s_n||^2, 0) \tag{1}$$

**Algorithm 1:** Imitation from observation with bootstrapped contrastive learning

$D_e = \{O_e^i\}_{1 \leq i \leq N} = \{(o_{e,0}^i, o_{e,1}^i, ..., o_{e,T}^i)\}_{1 \leq i \leq N}$
Initialize $f_\omega$, $g_\theta$, $q_\gamma$, $d_\phi$
`// Alignment Phase:  Training `$f$`, `$g$
**while** $k \leq N_{pretrain}$ **do**
   $\{O_e^i\}_{1 \leq i \leq n} \leftarrow sample(D_e)$
   $\{O^i\}_{1 \leq i \leq n} \leftarrow \pi_{random}(env)$
   Eval. $\mathcal{L}(\theta, \gamma, \omega, \phi)$ with $(\{O_e^i\}, \{O^i\})$
   Optimize $\theta, \gamma, \omega, \phi$
**end**
`// Interactive Phase:  Training `$\pi$`, `$Q$`, `$f$`, `$g$
Initialize $D_a = \{\}$
Initialize $\mathcal{D} = \{\}, Q, \pi$ `// Replay buffer, Q-value function, policy`
**while** $step \leq N_\pi$ **do**
   $O_e \leftarrow sample(D_e)$
   $o_0 \leftarrow env.reset()$
   **for** $t \in 0, .., T-1$ **do**
      $a_t \leftarrow \pi(o_t)$
      $o_{t+1} \leftarrow env.step(a_t)$
      $r_{t+1} \leftarrow -||f(o_{0:t+1}) - f(o_{e,0:t+1})||$
      Save $(o_t, a_t, o_{t+1}, r_{t+1})$ into $\mathcal{D}$
      $(o, a, o', r) \leftarrow sample(\mathcal{D})$
      Using DrQ-v2 update $Q, \pi$ with $(o, a, o', r)$
   **end**
   **if** $(step \leq N_{train})$ and
   $(step \mod N_{update} = 0)$ **then**
      Save $O = \{o_{0:t+1}\}$ into $D_a$
      $\{O_e^i\}_{1 \leq i \leq n} \leftarrow sample(D_e)$
      $\{O^i\}_{1 \leq i \leq n} \leftarrow sample(D_a)$
      Eval. $\mathcal{L}(\theta, \gamma, \omega, \phi)$ with $(\{O_e^i\}, \{O^i\})$
      Optimize $\theta, \gamma, \omega, \phi$
   **end**
**end**

where $s = g_\theta(o)$, $s_p = g_\theta(o_p)$. The training objective of the auto-encoder is then to minimize the loss function $\mathcal{L}_{ae}(\theta, \gamma) = ||o - q_\gamma(g_\theta(o))||^2$. Finally, we incorporate contrastive multiview coding to enrich the visual representations following [23]. Specifically we consider the color space transformation $\{R, G, B\} \rightarrow \{L, ab\}$ where: (1) the component $L$ is the view $v_1$, processed by the CNN $g_{\theta_1}$, (2) the component $ab$ is the view $v_2$ processed by $g_{\theta_2}$. The final encoding $s$ of an image $o$ is given by $s = g_\theta(o) = [g_{\theta_1}(v_1), g_{\theta_2}(v_2)]$. Following the formulation [23], we align the different views using a contrastive learning loss denoted $\mathcal{L}_S$. This process is illustrated in Figure 2. Overall, the objective function to be minimized for the training of the image encoding function is:

$$\mathcal{L}_{frame}(\theta, \gamma) = \mathcal{L}_S(\theta) + \mathcal{L}_{triplet}(\theta) + \mathcal{L}_{ae}(\theta, \gamma) \tag{2}$$

**Sequence encoding** In order to allow us to construct a reward function, we learn to represent a sequence of frames [7; 8] with a semantically meaningful distance between them. We use two losses: (1) $\mathcal{L}_Z$ a standard video representation learning approach Dense Predictive Coding (DPC) [8]; (2) and $\mathcal{L}_O$ bootstrapped expert behavior loss, which encourages separation between expert and non-expert trajectories. The loss, $\mathcal{L}_O$, lies at the heart of our method and allows the sequence representation to gradually improve as non-expert trajectories improve. DPC is a self-supervised learning method based on the surrogate task of predicting the next frame using the sequence of previous frames as input. Let us consider a dataset of videos $\mathcal{D} = \{O^i\}_{1 \leq i \leq N} = \{o_{0:T}^i\}_{1 \leq i \leq N}$. Denote, $d_\phi$, which from an encoding $z_t$ of a sequence of states $s_{0:t}$, predicts the next state $\hat{s}_{t+1}$ of the sequence. Following [8] we can use this for spatio-temporal representation learning. We first predict the next state $\hat{s}_{t+1} = d_\phi(f_\omega(s_1, \ldots, s_t))$ and subsequently use this to predict additional K-1 states $\hat{s}_{t+k} = d_\phi(f_\omega(s_1, \ldots, s_t, \hat{s}_{t+1}, \cdots, \hat{s}_{t+k-1}))$. We then apply the learning objective to minimize the

loss function $\mathcal{L}_Z$ given by:

$$\mathcal{L}_Z(\theta, \omega, \phi) = - \underset{o_{0:T}}{\mathbb{E}} \left[ \frac{1}{K} \sum_{1 \leq k \leq K} L_{t+k} \right] \tag{3}$$

$$L_t = \log \frac{\mathtt{h}(\hat{s}_t, s_t)}{\mathtt{h}(\hat{s}_t, s_t) + \sum_{t' \neq t} \mathtt{h}(\hat{s}_t, s_{t'})} \tag{4}$$

where $\mathtt{h}(a, b) = \exp(\frac{a^T b}{\tau \|a\| \|b\|})$ and $\tau$ is the temperature parameter. Indeed, the loss function is also a contrastive loss function where the positive pairs are $(\hat{s}_t, s_t)$ and the negative pairs are $(\hat{s}_t, s_{t'})_{t \neq t'}$ in the same demonstration. To bring sequences from the same distribution closer together, we use another contrastive loss that applies to the sequence encoding vectors $z$ returned by $f_\omega$. Thus, for a sequence $O$ belonging to one of the two trajectory distributions, we consider a positive sequence $O_p$ of the same distribution and $k$ negative $\{O_{n,i}\}_{1 \leq i \leq k}$ of different distributions. The objective function to be minimized is $\mathcal{L}_O$:

$$\mathcal{L}_O(\theta, \omega) = - \underset{(O, O_p, \{O_{n,i}\})}{\mathbb{E}} \left[ \log \frac{\mathtt{h}(z, z_p)}{\mathtt{h}(z, z_p) + \sum_i \mathtt{h}(z, z_{n,i})} \right] \tag{5}$$

$$z = f_\omega(g_\theta(o_0), \ldots, g_\theta(o_T)) \tag{6}$$

The objective function that we seek to minimize to train the sequence encoding function is $\mathcal{L}_{seq}(\theta, \omega, \phi) = \mathcal{L}_Z(\theta, \omega, \phi) + \mathcal{L}_O(\theta, \omega)$. Finally, for a triplet $(O, O_p, \{O_{n,i}\}_{1 \leq i \leq k})$, the frames used to evaluate $\mathcal{L}_{frame}$ come from the sequences of the sample, allowing us to compute the objective loss function $\mathcal{L}(\theta, \gamma, \omega, \phi) = \mathcal{L}_{frame}(\theta, \gamma) + \mathcal{L}_{seq}(\theta, \omega, \phi)$.

## 3 Experiments

We evaluate our method using a diverse set of continuous control tasks including 4 tasks (Reacher Hard, Finger Turn Easy, Hopper Stand, and Walker Run) from the DM Control Suite [22] and 3 tasks (Button Press, Plate Slide, Drawer Close) from the Meta-world environment [30]. The DM Control tasks require our agent to learn to coordinate multiple torque-controlled actuators. The Meta-world end-effector is controlled in a 3DOF task space with a 1DOF parallel jaw gripper modeled on a Sawyer Robot Arm. This environment setting requires our agent to learn to manipulate external objects with complex physical interactions with the world. In these manipulation tasks, we now have to model the robot and object-object interactions, such as in the case of Button Press and Drawer Close where the constrained object joint must be activated along a single axis. Below we describe the network architecture, details of our training procedures, and the results on these two datasets. All episode trajectories begin from randomized starting states of the robot and interactive objects (when applicable).

**Network architectures** Our image encoder function $g_\theta$ consists of a pair of convolutional neural networks $g_{\theta_1}$ and $g_{\theta_2}$. These models are of similar architectures, but the number of input channels is 1 for $g_{\theta_1}$ and 2 for $g_{\theta_2}$. We re-use the encoder architecture proposed by [17]. The architecture is a succession of four $5 \times 5$ stride-2 convolutional layers with 64, 128, 256, and 512 filters. These convolutional layers are followed by two $1 \times 1$ convolutional layers of 512 and 128 filters. All layers before the last are followed by BatchNorm [13] and a Leaky ReLU [9] activation function ($leak = 2$). The image decoding function $q_\gamma$ has an inverse architecture to the encoder, except that transposed convolutional layers are employed for the last 4 layers. Additional network details are presented in Tables 3 and 4 in Appendix A. The sequence encoder function, $f_\omega$, is a 2-layer LSTM with 128 as input and output sizes. The output sequence is followed by a fully connected layer with input and output size of 128. The next-state predictor is a sequence of two $128 \times 128$ fully connected layers with Leaky ReLU activation between the two layers. We employ an image encoding function, $g_\theta$, to learn a latent representation from high-dimensional observations. This function is trained with a self-supervised method that combines different surrogate objectives. However, much modern research [5; 28; 19] has shown the benefit of using pre-trained or foundation models trained on a large corpus for downstream tasks. We illustrate the opportunity of using this strategy with our setup by employing EfficientNet [23], a convolutional model trained on ImageNet, as a backbone image encoder. In this

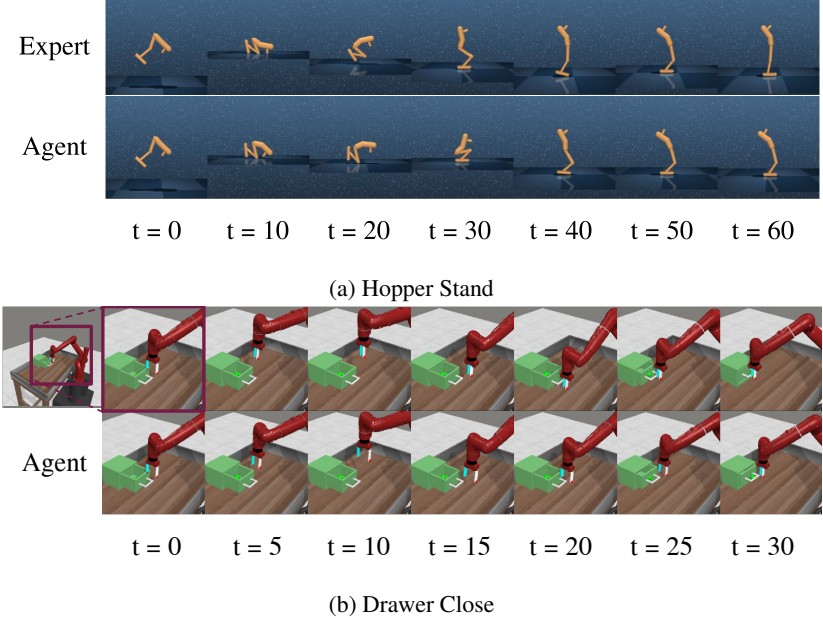

(a) Hopper Stand

(b) Drawer Close

Figure 3: Comparison of the actions taken between an expert (top row) policy and imitation agent (bottom) learned using our proposal. We show learned agents in Hopper Stand and Drawer Close with the same initial conditions. Observe that for Hopper Stand the agent behavior of our learned agent is very similar to that of an expert. For Drawer Close although the learned agent takes a different trajectory than the expert(e.g. keeping the gripper wider open) it is able to solve the task.

setting, the function $g_\theta$ is composed of the backbone model with the weights frozen followed by a 3-layer multilayer perceptron (MLP) of size $1280 \times 512 \times 128$ with trainable parameters. Though we utilize EfficientNet-B0 as a backbone, there are many reasonable choices for pre-trained encoders.

**Training: Alignment Phase**  For each task, we build a dataset of expert episodic trajectories with colored image observations at each timestep by running a trained RL agent and rendering visual images of the environment as needed. We also build a second dataset of trajectories with a randomly defined policy using the same process. In this work, we employ DrQ-v2 agents as our experts for imitation, though any reasonable expert (image-based RL, planning agents, human, or animal experts) could be used since we do not need access to the state of the agent. In the case, where we train an image encoder from scratch (BootIfOL), the demonstration datasets $\mathcal{D}_e$ and $\mathcal{D}_a$, of size $5000 \times 51 \times 64 \times 64$ (trajectory $\times$ timestep $\times$ height $\times$ width), are used as input. In Eff-BootIfOL, where we leverage large-scale pre-training, we utilize smaller datasets of demonstrations for finetuning: $1500 \times 61 \times 224 \times 224$ (trajectory $\times$ timestep $\times$ height $\times$ width).

**Training: Interactive Phase**  The parameters of the encoding functions are updated during $N_{\text{train}} = 375K$ training steps of the Interactive Phase. After these training steps, the parameters of the encoding functions are frozen. Our RL agent training is performed over $N_\pi = 1550K$ training steps. We employ DrQ-v2 [29], an image-based RL agent based off of the DDPG [16] architecture, as our RL policy learner with default hyperparameters. Additional hyperparameters are available in Table 7.

**General results**  We perform the training on the Reacher Hard, Finger Turn Easy, Hopper Stand, and Walker Run tasks from the Deepmind Control Suite [22]. We compare our algorithm to the Context Translation (CT) [17] method for IfO by varying the number $n$ of expert trajectory samples used at each training episode to predict the CT agent trajectory. We also baseline against ViRL [1] and GAIfO [25]. Demonstrated across 4 tasks in DM Control, our method shows strong performance in learning to complete tasks given visual demonstrations as shown in Table 1, where GAIfO showed poor performance on all tasks. To facilitate the comparison, we have scaled the average returns between 0 and 1 in Table 1. The expert has the highest average return, one, and the average return of the random agent is zero. We present the non-scaled absolute values in Appendix B. We also show promising results in complex visual scenes by employing a pre-trained backbone on Meta-world environments as shown in Table 2. Average episodic returns over RL training in the Interactive Phase

Table 1: Evaluation of the average return over 500-step episodes of agents trained with the Context translation (CT) [17] and ViRL [1] algorithms. We evaluate the agents on the Reacher Hard, Finger Turn Easy, Hopper Stand, and Walker Run tasks. For 3 of the environments our approach exceeds the existing methods by a wide margin. For Reacher Hard we are able to achieve rewards on par with the Expert policy, while our comparison methods completely fail to learn good policies. Though this is a fairly simple control problem (a visual version of inverse kinematics), the distribution of starting states and goals is fairly large compared to other tasks we look at. Our technique of training with failure demonstrations is particularly advantageous in this setting as we see more of the state space. Returns in this table are scaled so that the average expert return is one and the average random return is zero.

| Agent | Avg. return | | | |
| --- | --- | --- | --- | --- |
| | Reacher Hard | Finger turn easy | Hopper Stand | Walker Run |
| BootIfOL (ours) | **0.99** | 0.13 | **0.74** | **0.06** |
| ViRL | 0.0 | 0.08 | 0.58 | 0.0 |
| CT (n=10) | 0.10 | **0.18** | 0.0 | 0.0 |
| CT (n=1) | 0.04 | 0.13 | 0.0 | 0.03 |

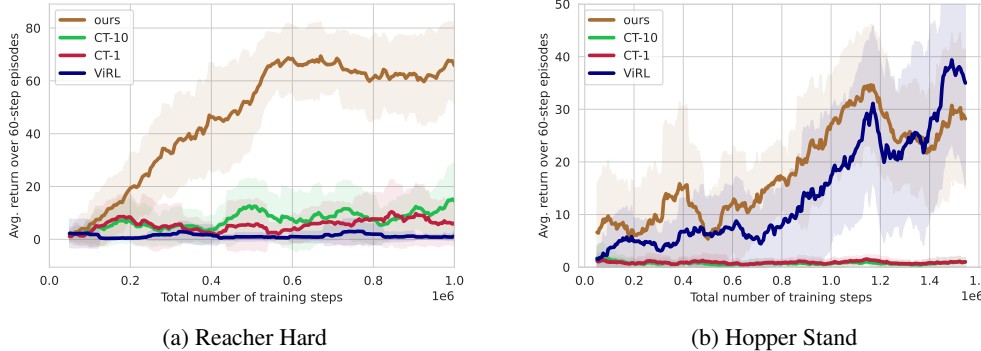

(a) Reacher Hard                                  (b) Hopper Stand

Figure 4: Average returns throughout agent training in Interactive Phase compared to CT and ViRL agents on Reacher Hard and Hopper Stand tasks. We observe that agent progresses quickly in both cases as compared to other baselines. Although ViRL is able to do well in Hopper Stand, it is unable to tackle all environments (e.g. Reacher Hard).

are presented in Figure 4. We show visual examples of our agents acting in Hopper Stand and Drawer Close tasks in Figure 3. We present the examples for other tasks in Figure 9 and describe them in Appendix E. We observe that in tasks such as Walker Run, where a great degree of coordination between joints is necessary to control the robot, IfO agents struggle to reach even $10\%$ of the expert's reward (see Table 1).

Table 2: Evaluation of the average return over 500-step episodes of our agent (*BootIfOL*) where the encoder was trained from scratch and an agent which exploited EfficientNet-B0 [19] as a backbone model (*Eff-BootIfOL*). We evaluate these agents on the manipulation tasks: Button Press, Plate Slide, and Drawer Close in the Meta-world simulator [30]. The returns are scaled so that the average expert return is one and the average random return is zero. We observe that on this challenging visual environment a pre-trained image representation is needed to obtain maximum performance. We note other baselines are unable to tackle this environment, while our method is able to obtain close to expert performance in some tasks (Button Press and Drawer Close) and non-trivial performance in others (Plate Slide). We present the non-scaled absolute values in Table 6 in Appendix B.

| Agent | Avg. return | | |
| --- | --- | --- | --- |
| | Button Press | Plate Slide | Drawer Close |
| Eff-BootIfOL (ours) | **0.70** | 0.25 | **0.91** |
| BootIfOL (ours) | 0.0 | 0.25 | 0.53 |

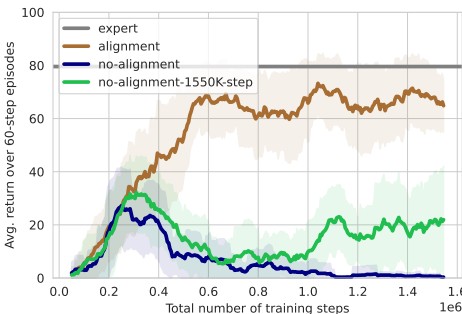

Figure 5: Ablating the effect of the encoding functions' training in the Alignment Phase (described in Section 2) on the final performance of the agent on the Reacher Hard task. *alignment* is our initial model with the Alignment Phase executed; *no-alignment* and *no-alignment-1550K-step* are the models without execution of the Alignment Phase. In *no-alignment-1550K-step*, the encoding functions are updated continuously until the end of the agent's training.

**Results of Evaluation on Meta-World**    Considering the Meta-world environment [30], we study precisely the importance of a pre-trained convolutional network as a feature extraction model. The visual complexity offered by Meta-world tasks allows us to evaluate the relevance of an accurate image encoder during the training of the encoding functions. We evaluate our new agent, trained with a learned distance metric using a pre-trained image encoding, as described in the Network Architectures section ( 3), on the tasks Button Press, Drawer Close, and Plate Slide of the Meta-world environment. In Table 2, we compare the average return of this agent with our initial agent. The results in Table 2 demonstrate the importance of the image encoding function for the effectiveness of the learned distance metric. In particular, these results show that the success of the reward function learning depends on the level of accuracy in the interpretation of each frame. We observe a particularly weak result concerning the Plate Slide task. We hypothesize that this failure is due to a difficulty related to the sequence encoding function, which has a different role. Despite the accuracy of the feature extractor, the task remains difficult to imitate by our agent.

**Ablating the Alignment Phase**    In our algorithm, we integrate the Alignment Phase, described in Section 2, allowing us to bootstrap the encoding functions $f$ and $g$ on two trajectory distributions: (1) Expert's trajectories, (2) random trajectories. This particularity is not present in most methods such as GAN-like approaches. We hypothesize that providing, from the beginning of the agent's training, significant and consistent rewards offers an important gain on the search for the optimal imitation policy. We evaluate the importance of this step by removing the Alignment Phase in two ways: (1) the encoding functions are trained during 375K total training steps (standard case) in the Interactive Phase; (2) the encoding functions are trained during all the agent's training in the Interactive Phase (similarly to ViRL and GAIfO). We present the results of this experiment in Figure 5. We observe that the use of the alignment phase leads to drastically better performance, highlighting this as a critical phase. Although training these for the full interactive phase can lead to some progress, it is not nearly as efficient as the bootstrapped approach that includes the Alignment Phase.

**Ablating the use of a Learned Image-Encoder for RL**    To date, in our experiments, the RL agent is learned through a dissociate policy function $\pi$ and q-value function $Q$ as proposed [29]. In this architecture, the policy function $\pi$ is composed of a CNN $E$ that internally encodes the observation images, followed by an MLP that feeds the image encodings to return actions. $E$ is shared between the policy $pi$ and the associated q value function $Q$, which feeds encoding-action pairs to return q-values. A natural question is whether we can directly link our learned image encoding function $g_\theta$ to $pi$ and $Q$ by dropping the CNN $E$. The interest of this approach is to transfer the learning of the $g$ to the agent in order to learn directly the extracted features. We know that in modern RL approaches, using low-dimensional state vectors generally improves agent performance and efficiency, because of higher precision in the description of the state. We are interested in learning if our learned observation feature descriptions - optimized with a pixel-based reconstruction loss - provide enough information about the state to enable an RL policy to learn efficient and quality control policies without needing access to either the true simulator state or raw pixels. Specifically, we replace $E$ with an MLP shared between $\pi$ and $Q$. The image encodings returned by $g$ are the input data of the MLP $E$. Note in this experiment we freeze the parameters of $g_\theta$ with respect to the RL update and call this ablation the

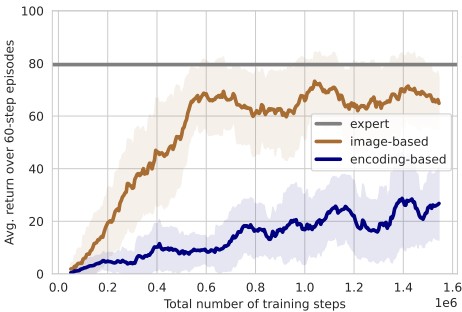

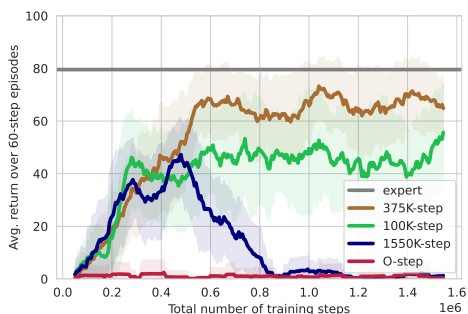

| (a) Evaluation of the Encoding-Based Agent | (b) Evaluation of the encoder training duration |

Figure 6: Ablation studies. We show the average return of the agent on the Reacher Hard task over 5 episodes of 60 steps. On the left in 6a We evaluate whether we can re-use the image encoding CNN from our imitation function for policy learning (encoding-based) or whether the RL agent should optimize a new image encoding network (image-based). We observe that attempting to use the encodings from BootIfOL directly in the policy network (encoding-based) degrades performance. On the right 6b we evaluate the agent with respect to the training duration of the encoding functions. If they are not trained after the Alignment Phase (*0-step*), the rewards are non-informative. Similarly, if we continue to train them as the agent begins to converge to a strong policy, they can degrade the reward signal.

*encoding-based agent*. We present in Figure 6a the average return of the original image-based agent and the encoding-based agent on the Reacher Hard task over training. This leads us to believe that the image encodings returned by $g$ present biases that limit the understanding of the environment by the policy, thus preventing this policy from achieving performance similar to agents with access to image-based states directly. We suppose that this bias is because $g$ and $f$ are trained jointly with learning objectives that are not optimized for the policy function.

**Effect of Encoder Training Length in Interactive Phase**   In the *Interactive Phase*, we train the trajectory and image encodings for $N_{train}$ steps before switching to only using a standard RL algorithm. We now study how the length of this phase affects the final reward reached by the agent. We evaluated our algorithm with $N_{train}$ set to a period of 0, 100K, 375K, and 1550K total training steps. Updates to encoder parameters happen every $N_{update} = 50$ steps. The results of these experiments on the Reacher Hard task are presented in Figure 6b. We observe that when the encoding functions are not updated during the Interactive Phase, learning does not proceed, as the rewards provided by the Alignment Phase are not sufficient. Increasing the number of $N_{train}$ steps initially has a positive effect on average returns per episode, reaching close to the expert return per episode in the case of $N_{train} = 375K$ steps. However, when the encoding functions are not frozen training during the Interactive Phase, the learned reward function becomes brittle due to changing state encoding. Thus we observe that $N_{train} = 1550K$ steps eventually led to a collapse in the policy.

## 4   Conclusion

We present a new method of Imitation from Observation that relies on the encoding of agent behavior using exemplar demonstrations from visual observations. This method uses self-supervised learning of states and sequences to progressively train a reward function for use by a reinforcement learning agent. We demonstrate the strength of this method on a set of 7 simulated robotic tasks which have access to a limited set of expert demonstrations. We note that our adopted problem framework: imitation from image observations, though important for its potential practical applications, is challenging from both the computational and observational perspectives for tasks that require precise control. In this paper, we explore this challenge by training our agents with access to information from task-specific and pre-trained image encoders and show that the fidelity of the encoding function is critically important for downstream control tasks. Our approach to increase the performance of the learned representation is to keep the data generated by the agent during the Interactive Phase. In future work, we see expanding the sequence optimization functions so that they evaluate not just the two agent trajectory distributions we have now (expert and non-expert), but distributions of many performance *levels* which are generated as the agent trains.

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

# A  Neural network architectures

Table 3: Architecture of convolutional neural networks $g_{\theta_1}$ and $g_{theta_2}$. $C$ is the number of channels of the output matrix at each layer. $K$ is the size of the convolution kernel at each layer. $S$ is the stride of the convolution operation. $P$ is the padding added initially to the input matrix.

| Encoder network | | | | |
| --- | --- | --- | --- | --- |
| Layer | C | K | S | P |
| Image ($64 \times 64$) | 1,2 | - | - | - |
| Conv + BNorm + LReLU | 64 | 5 | 2 | 0 |
| Conv + BNorm + LReLU | 128 | 5 | 2 | 0 |
| Conv + BNorm + LReLU | 256 | 5 | 2 | 0 |
| Conv + BNorm + LReLU | 512 | 5 | 2 | 0 |
| Conv + BNorm + LReLU | 512 | 1 | 1 | 0 |
| Conv | 128 | 1 | 1 | 0 |

Table 4: Architecture of the image decoding function $q_\gamma$ based on transposed convolution operations. $C$ is the number of channels of the output matrix at each layer. $K$ is the size of the convolution kernel at each layer. $S$ is the stride of the convolution operation. $OP$ is the padding added to the output matrix. The padding added to the input matrix is always 0.

| Decoder network | | | | |
| --- | --- | --- | --- | --- |
| Layer | C | K | S | OP |
| Image encoding ($1 \times 1$) | 128 | - | - | |
| Conv + LReLU | 512 | 1 | 1 | 0 |
| Conv + BNorm + LReLU | 512 | 1 | 1 | 0 |
| TConv + BNorm + LReLU | 256 | 5 | 2 | 0 |
| TConv + BNorm + LReLU | 128 | 5 | 2 | 0 |
| TConv + BNorm + LReLU | 64 | 5 | 2 | 1 |
| TConv | 3 | 5 | 2 | 1 |

# B  Estimation of our models and baselines with absolute average episodic return

We detail in Tables 5 and 6 the non-scaled average episodic returns of our models and baselines on each of the 7 tasks of the experiments.

# C  Encoding function losses

During the alignment phase, we evaluate the intermediary loss terms $\mathcal{L}_Z$, $\mathcal{L}_{seq}$, $\mathcal{L}_{ae}$ and $\mathcal{L}_{triplet}$, on the expert trajectories and random trajectories. These different functions are used to bootstrap the encoding functions for the next phase and each have a well-defined learning objective as explained in Section 2. We detail the evolution of these loss terms during the alignment phase in Figure 8. In the interactive phase, during the training of the imitation agent, the encoding functions are progressively

Table 5: Evaluation of the average return over 500-step episodes of agents trained with the Context translation (CT) [17] and ViRL [1] algorithms. We evaluate the agents on the Reacher Hard, Finger Turn Easy, Hopper Stand, and Walker Run tasks.

| Agent | Avg. return | | | |
| --- | --- | --- | --- | --- |
| | Reacher Hard | Finger turn easy | Hopper Stand | Walker Run |
| Expert | $850.52 \pm 315.70$ | $893.44 \pm 219.75$ | $880.93 \pm 76.89$ | $789.77 \pm 17.25$ |
| BootIfOL (ours) | $\mathbf{843.16 \pm 274.84}$ | $199.76 \pm 399.02$ | $\mathbf{651.33 \pm 362.95}$ | $\mathbf{79.50 \pm 1.25}$ |
| ViRL | $0.44 \pm 1.27$ | $160.2 \pm 366.52$ | $516.27 \pm 417.21$ | $29.31 \pm 24.72$ |
| CT (n=10) | $97.84 \pm 254.52$ | $\mathbf{238.92 \pm 422.07}$ | $1.42 \pm 3.22$ | $25.15 \pm 18.71$ |
| CT (n=1) | $38.04 \pm 127.53$ | $199.48 \pm 396.99$ | $1.01 \pm 1.71$ | $52.42 \pm 34.28$ |
| Random | $6.64 \pm 16.94$ | $92.6 \pm 202.68$ | $2.44 \pm 5.19$ | $29.93 \pm 4.52$ |

Table 6: Evaluation of the average return over 500-step episodes of our agent (*BootIfOL*) where the encoder was trained from scratch and an agent which exploited EfficientNet-B [19]0 as a backbone model (*Eff-BootIfOL*). We evaluate these agents on the manipulation tasks: Button Press, Plate Slide, and Drawer Close in the Meta-world simulator [30].

| Agent | Avg. return | | |
| --- | --- | --- | --- |
| | Button Press | Plate Slide | Drawer Close |
| Expert | $556.06 \pm 6.46$ | $734.56 \pm 172.51$ | $4223.65 \pm 21.14$ |
| Eff-BootIfOL (ours) | $\mathbf{434.39 \pm 139.74}$ | $340.45 \pm 46.56$ | $\mathbf{3949.90 \pm 681.42}$ |
| BootIfOL (ours) | $146.56 \pm 17.71$ | $\mathbf{342.40 \pm 93.08}$ | $2847.10 \pm 1679.56$ |
| Random | $153.29 \pm 51.68$ | $209.12 \pm 87.38$ | $409.07 \pm 1288.89$ |

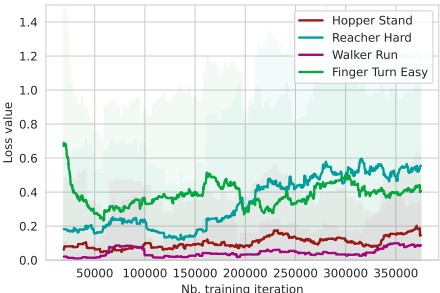

Figure 7: Loss term $\mathcal{L}_{seq}$ throughout the Interactive Phase during which the agent is trained as described in Section 2. The effect of the RL agent learning in the environment is seen in this plot.

trained on the new trajectories of the agent and on the initial expert trajectories. We present in the Figure 7, the evolution of $\mathcal{L}_{seq}$ loss function. We can see a slight tendency for $\mathcal{L}_{seq}$ to increase. This effect is due to the sequence encoding function, which is increasingly unable to distinguish expert sequences from agent sequences.

## D  Hyperparameters

Hyperparameters of our method and the default values are presented in Table 7.

## E  Demonstrations

We present in Figure 9, examples of the execution of each of the tasks by the expert and the agent, under the same initial conditions.

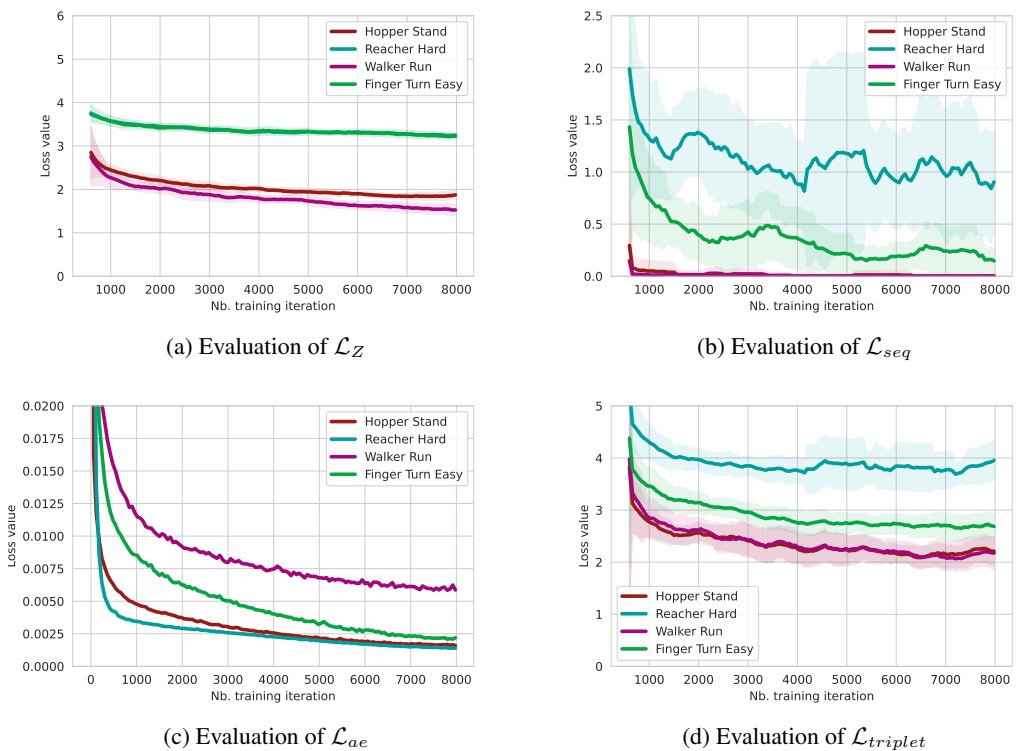

Figure 8: Evolution of the loss terms $\mathcal{L}_Z$, $\mathcal{L}_{seq}$, $\mathcal{L}_{ae}$ and $\mathcal{L}_{triplet}$ during the training of the encoding functions $g_\theta$ and $f_\omega$ in the Alignment Phase.

Table 7: Hyperparameters

| Parameter | Value |
|---|---|
| Number of expert trajectories (BootIfOL) ($N$) | 5000 |
| Number of expert trajectories (Eff-BootIfOL) ($N$) | 1500 |
| Number of frames in each trajectory (BootIfOL) ($T$) | 61 |
| Number of frames in each trajectory (Eff-BootIfOL) ($T$) | 51 |
| Size of images (BootIfOL) | $64 \times 64$ |
| Size of images (Eff-BootIfOL) | $224 \times 224$ |
| Number epochs during the Alignment Phase ($N_{pretrain}$) | 8000 |
| Number of videos per batch ($n$) | $16 \times 2$ (pairs) |
| Total number of training steps during which encoders are trained ($N_{train}$) | 375K |
| Periodicity of encoder parameter update (in training step) ($N_{update}$) | 50 |
| Total number of agent training steps ($N_\pi$) | 1.55 M |
| Learning rate | $1 \times 10^{-4}$ |
| Optimizer | Adam [15] |

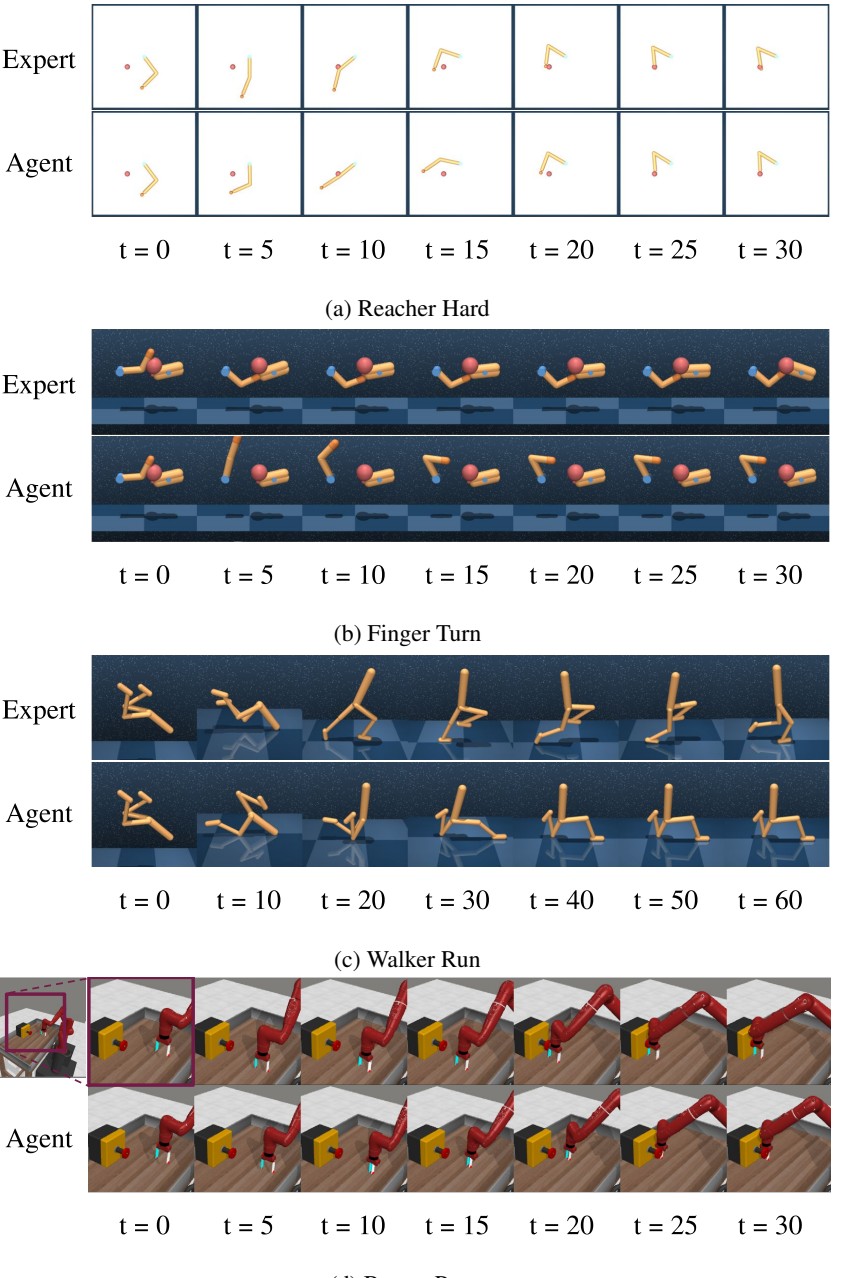

Figure 9: Comparison of the actions between expert and agent in each task and in the same initial conditions.

