# OpenReview forum: "Imitation from Observation With Bootstrapped Contrastive Learning"
_NeurIPS.cc/2022/Workshop/Offline_RL — Offline RL Workshop NeurIPS 2022_

### Official Review · Reviewer_ccKw · 2022-10-19

**Rating:** 6
**Confidence:** 2

**Review:**

This paper proposes a technique for learning a reward function from given expert demonstrations, and subsequently trains a policy using online RL. The paper has some interesting experiments in this setting on DM-control tasks.

In general, I came out of this paper unsure about the technical contribution of the work. The general recipe of this type of adversarial reward training scheme has been done many times before (e.g. AIRL and successive variants) -- what exactly is new in what the authors propose. I also found the experimental section somewhat underwhelming, and think it could be strengthened by better ablatory studies and a wider set of domains beyond visual DM-control tasks.

Nonetheless, I think this is an interesting avenue of research and worth discussing at this workshop.

---

### Official Review · Reviewer_hcd1 · 2022-10-20

**Rating:** 6
**Confidence:** 4

**Review:**

This paper studies the problem of Imitation from Observation, which involves an agent learning to solve a task by observing only the states visited by an expert that can solve the task. This variation of imitation learning is hard because without actions, the learning algorithm must build an internal model that relates its own actions to the expert observations. This paper proposes to build such a model by learning a latent representation of video trajectories with contrastive learning, and using the inner product between contrastive representations as a reward. Results on a handful of visual continuous control tasks from dm-control suggest the proposed method outperforms other methods for Imitation from Observation. Overall, the results in the paper are promising, but certain important ablations are missing. I recommend the paper be accepted to the workshop, but I also encourage the authors to consider these questions and suggestions:

Questions:

(1) There are several terms in the loss function used to learn the latent representation. How are these terms balanced?

(2) How do the five terms in the loss function relate in terms of their relative importance and contribution to final performance?

(3) What is the architecture of each of the constituent models (g and f)?

(4) What is the explanation for why the image-based agent outperforms the state-based agent in Figure 2? This was surprising on first read since this behavior is typically reversed in other papers and is worth explaining. If this experiment is not testing with proprioceptive state, perhaps another label to replace "state-based" in Figure 2b would be helpful to avoid confusing the reader.

Suggestions:

(1) On lines 118-119 consider changing "long Short-term memory" to Long Short-Term Memory (LSMT) and citing Hochreiter & Schmidhuber, 1997. Also, in Table 1, Ours + Walker Run is likely supposed to read "794.98 ± 1.25" but instead reads "79.498 ± 1.25".

I recommend correcting other grammatical mistakes beyond the ones listed above as well.